# The Concentration of Pro- and Antiangiogenic Factors in Saliva and Gingival Crevicular Fluid Compared to Plasma in Patients with Peripheral Artery Disease and Type 2 Diabetes

**DOI:** 10.3390/biomedicines11061596

**Published:** 2023-05-31

**Authors:** Iwona Gregorczyk-Maga, Aleksandra Szustkiewicz-Karoń, Mateusz Gajda, Maria Kapusta, Wojciech Maga, Martyna Schönborn

**Affiliations:** 1Faculty of Medicine, Institute of Dentistry, Jagiellonian University Medical College, 31-155 Krakow, Poland; 2Department of Angiology, Faculty of Medicine, Jagiellonian University Medical College, 31-121 Krakow, Poland; 3Doctoral School of Medicine and Health Sciences, Jagiellonian University Medical College, 31-121 Krakow, Poland; 4Department of Clinical Biochemistry, Jagiellonian University Medical College, 30-688 Krakow, Poland

**Keywords:** gingival crevicular fluid, angiogenesis, circulating proangiogenic factor, circulating antiangiogenic factor, PAD

## Abstract

Several studies have investigated various biomarkers in relation to peripheral artery disease (PAD) for disease stratification and early-onset detection. In PAD, angiogenesis is required for tissue restoration and tissue perfusion. Considering changes in angiogenesis in patients with PAD, angiogenic factors could be explored as one of the new prognostic molecules. In recent studies, saliva and gingival crevicular fluid (GCF) have gained recognition as new, easily obtained diagnostic materials. This study aimed to compare the levels of selected circulating angiogenic factors (VEGF-A, PDGF-BB, and ANG-1) in unstimulated whole saliva (WS) and GCF versus plasma at three points in time to find possible correlations between their concentrations among patients with PAD and diabetes type 2 in 32 patients with Rutherford stages 5 and 6. A significant positive correlation has been demonstrated between circulating PDGF-BB levels in GCF and plasma. In most cases, comorbidities do not have an impact on the change in general correlation for the whole group. Our results clearly showed that GCF could be a good source for PDGF assessment. However, future studies with a larger number of subjects are warranted to confirm this finding and identify the most accurate angiogenic biomarkers in saliva or GCF that could be applied in clinical practice.

## 1. Introduction

In peripheral arterial disease (PAD), due to ischemia resulting from inadequate blood supply, structural alterations occur in existing blood vessels or the formation of new vessels begins. However, the capacity of the processes is limited, and restoration of limb perfusion to normal levels in PAD patients is usually impossible [1]. Angiogenesis, also called neovascularization, is the formation of new vessels from existing vascular structures and is required for tissue repair, remodeling, and restoration of tissue perfusion [2]. Numerous studies have presented the role of physiologically produced substances that are known to stimulate angiogenesis, such as VEGF (vascular endothelial growth factor) and its isoform VEGF-A, which was presented mainly in wounds, or PDGF (platelet-derived growth factor) [3,4]. On the other hand, there are also well-described substances with antiangiogenic characteristics, such as Ang-1 (angiopoietin-1), a substantial and potent maturation marker that stabilizes endothelial cells in capillaries [5,6]. Several studies have investigated various biomarkers in plasma in relation to PAD [7]. Some inflammatory molecules, such as C-reactive protein (CRP), a few types of interleukins, matrix metalloproteinases (MMPs), extracellular vesicles (EVs), and non-coding RNAs, are explored for use in PAD identification and outcome evaluation [8,9].

Considering the global burden of PAD (over 200 million people worldwide) [10] and the risk of complications such as nonhealing ulcers, limb amputations, and, as a result, an increased risk of death, there is still a need to find reliable biomarkers for disease stratification and early onset detection. We hypothesize that, considering changes in angiogenesis in patients with PAD, angiogenic factors could be explored as new biomarkers.

In recent studies, saliva and gingival crevicular fluid (GCF) have gained recognition as new, easily obtained diagnostic materials. Saliva is a biofluid primarily composed of secretions from the major and minor salivary glands. However, it also contains components derived from the mucosal surfaces, gingival crevices, and tooth surfaces of the mouth. It is a plasma ultrafiltrate comprising proteins originating both from the blood and salivary glands [11]. Another type of oral fluid that has been less commonly utilized in research and diagnosis is GCF, a serum exudate formed from the periodontal pockets. Its production is quite different and begins in the gingival capillaries. Pressure exerted by hydrostatic and osmotic forces in the capillaries of the gingiva leads to the movement of fluid into the interstitial space, which is subsequently drained by lymphatic vessels. The excess fluid collected by the lymphatic system escapes into the gingival crevice to form GCF. This theory is supported by the fact that small amounts of fluid drained from the gingival pocket have protein concentrations very similar to those of interstitial fluid. The amount of fluid produced in the gingival sulcus of a tooth ranges from 2.4–6 μL/h, depending on the tooth. GCF can be collected by three methods: lavage of the interdental space, suction using micropipettes, or strips of cellulose dedicated to this purpose. Despite some discrepancies among authors regarding the time of suction and the depth of insertion of the strip into the pocket, this method is considered one of the most preferred, simple, and efficient techniques for material collection.

GCF composition is similar to normal serum and includes immunoglobulins, peptides, proteins, enzymes, tissue degradation products, and microorganisms [12]. Immune system cells are primarily associated with active transportation, while the concentrations of other components indicate passive filtration that depends on the size of the molecule [1,13]. Biomarkers in saliva are derived from serum, mucosal transudate, and GCF. As diagnostic material, its recent development comprises point-of-care devices, rapid tests, or more standardized formats [14]. GCF and salivary material may have significant diagnostic value due to their potential positive correlation with blood. Furthermore, through its simple and non-invasive collection, there is growing interest in diagnosis based on its analysis.

This study aimed to compare the levels of selected circulating angiogenic factors in samples of unstimulated whole saliva (WS) and GCF against plasma in three-point time, as well as to find potential correlations between their concentrations among patients with PAD and diabetes type 2 (DM2). Furthermore, we attempted to answer whether patients’ comorbidities had an impact on the observed associations between measurements.

## 2. Materials and Methods

The analysis included 32 patients hospitalized in 2022 in the Clinical Department of Angiology. All patients had critical limb ischemia (CLI) with stages 5 or 6 in the Rutherford classification and concomitant DM2. Patients over 40 years of age with signed consent were enrolled in the study. The exclusion criteria comprised any active inflammation in the oral cavity or systemic infection.

Information on age, sex, chronic disorders, and smoking status was retrospectively extracted from medical documentation. Data on comorbidities, hemoglobin concentration, white blood cells, c-reactive protein, and plasma concentration of glycated hemoglobin (HbA1c) were evaluated at the beginning of the study. Additionally, extra blood samples, gingival crevicular fluid (GCF), and unstimulated saliva (WS) were collected from all patients to assess the chosen circulating pro- and antiangiogenic factors. The analysis was carried out at three points in time: a short period break (24 h—based on the T_1/2_ time of the factors) and a long period break (30 days—during routine follow-up after hospitalization).

Among factors with well-documented pro- and antiangiogenic effects, two proangiogenic factors (platelet-derived growth factor BB—PDGF-BB and vascular endothelial growth factor A—VEGF-A) and one antiangiogenic factor (angiopoietin-1—ANG-1) were selected to examine the preliminary tendency for changes and correlations.

Plasma was separated from the whole fasting blood with EDTA and centrifuged at 3000 rpm for 15 min. Pocket fluid was collected using paper strips (Periopaper, Oraflow Inc., Plainview, NY, USA). For this purpose, a paper strip was placed in the gingival pocket until a slight resistance was felt and held for 40 s. To avoid contamination of the GCF by saliva, the tooth surface was dried, and excess saliva was removed from the mucous membranes with sterile gauze. Before sampling, oral health was assessed visually to exclude the presence of any inflammation. Using the periodontal probe, the presence of calculus and bleeding was excluded, and the depth of periodontal pockets was examined, which in the study group did not exceed the normal value of 3 mm. Unstimulated saliva samples (spitted after a fasting period) were collected in 10 mL plastic tubes. All materials were stored at −80 °C until the final analysis.

The concentration of VEGF-A was measured using the enzyme-linked immunosorbent assay (Human VEGF-A ELISA Kit; Thermo Fisher Scientific, Inc., Wilmington, NC, USA). Human ANG-1 and PDGF-BB levels were measured by the quantitative sandwich enzyme immunoassay technique (Human Angiopoietin-1 Quantikine ELISA Kit and Human PDGF-BB Quantikine ELISA Kit; R&D Systems, Minneapolis, MN, USA), according to the instructions provided by the manufacturers. Plasma requires a 1:2 dilution for VEGF-A, 1:15 for ANG-1, and 1:5 for PDGF-BB. Saliva requires 1:20 dilution for VEGF-A and ANG-1 and an undiluted sample for PDGF-BB. GCF requires a 1:2 dilution for VEGF-A and an undiluted sample for ANG-1 and PDGF-BB. Assay sensitivity was 7.9 pg/mL for VEGF-A, 3.45 pg/mL for ANG-1, and 7.5 for PDGF-BB. The intra-assay and inter-assay coefficients of variation [CV%] were as follows: 6.2% and 4.3% for VEGF-A; 2.7% and 5.8% for ANG-1; and 3.1% and 7.8% for PDGF-BB. Optical density was measured on a plate reader ELx808™ (Bio-Tek Instruments, Winooski, VT, USA) at the wavelength 450 nm, and data were collected using Gen 5 (Bio-Tek, New York, NY, USA) software. A four-parametric logistic (4-PL) curve fit was used to generate the standard curve. The results of the VEGF-A, ANG-1, and PDGF-BB measurements were expressed as pg/mL and recalculated for ng/mL due to a range of values.

To assess chronic limb ischemia, the Rutherford classification was included, which comprises five grades for minor tissue loss and six grades for major tissue loss extending above the transmetatarsal level [15]. The ankle-brachial index and toe-brachial index were used to assess the baseline level of ischemia related to a duplex ultrasound examination.

### 2.1. Statistical Analysis

Parameter analysis at time points was performed using the Friedman test, followed by pairwise analysis for statistically significant results with Bonferroni correction. The correlation analysis was performed using the Spearman correlation. For quantitative variables, the median, 25th, and 75th percentiles were used. For the description of the nominal variables, numbers and percentages were used. Statistical analysis was performed with IBM SPSS Statistics ver. 28.0.1.0. (IBM Corp, Armonk, NY, USA).

### 2.2. Ethics

The study was carried out according to the Declaration of Helsinki guidelines and was approved by the Jagiellonian University Bioethical Committee (No. 1072.6120.129.2020).

## 3. Results

Thirty-two patients were enrolled, most of whom were men (84.4%) with Rutherford stage 5 (59.4%). All patients were assumed to have type 2 diabetes. The most common comorbidity was hypertension, followed by dyslipidemia and coronary artery disease. More than 75% of the people had a positive history of nicotine (past or present). Considering gender, the mean hemoglobin level indicated mild anemia. The ABI and TBI indices, according to the mean values, were below the correct values (0.9 and 0.7, respectively). Up to 60% of the patients had elevated CRP (>5 mg/L), and 50% had leukocytosis greater than 10,000/μL. Detailed data on the study population are summarized in Table 1.

In our study, a time-dependent growth was observed for the PDGF-BB factor in plasma and GCF, which in Spearman’s correlation analysis showed a significant (*p*-value < 0.017), positive, and moderate correlation, as shown in Figure 1.

In the analysis of the correlation of pro- and antiangiogenic factors with the level of glycated hemoglobin, a significant negative correlation was obtained at two of three time points (0 and 30 days). The obtained correlation was on the border of moderate and weak correlations.

For the VEGF-A factor, including the medians in the three-point time analysis, the changes were similar in time in WS and GCF, as well as for the ANG-1 factor in WS and plasma (decreased on day 1 and increased on day 30) (Table 2). For individual factors, in addition to a comparison between the materials examined, an analysis of changes over time was also performed. Statistically significant changes were obtained for VEGF-A and ANG-1, which were then analyzed in pairs to determine significant differences within the group. Based on the analysis conducted, statistically significant differences were found for VEGF-A determined in saliva between the measurements at time points A (median 1803.60) and B (median 1501.40), resulting in a decrease in the average concentration, and between the measurements at time points B (median 1501.40) and C (median 2009.81), resulting in an increase in the average concentration. ANG-1 showed an increase at time point C (median 3105.96) relative to time point A (median 2879.01). Then an analysis was carried out, considering comorbidities. Significant changes in time for VEGF-A occurred in the presence of all comorbidities, except for atrial fibrillation and chronic renal failure, where the relationship was reversed (significant changes in VEGF-A in people without these diagnoses). In contrast, for the ANG-1 factor, the relationship was observed only in the case of a state after a heart attack and chronic coronary artery disease and in the group without atrial fibrillation and dyslipidemia. In the remaining diseases, significant variability disappeared for ANG-1.

There were no statistically significant changes in the concentration of pro- and antiangiogenic circulating factors considering the division of groups by age, sex, Rutherford classification, or indicators of limb ischemia such as ABI and TBI.

Due to the evaluation including comorbidities for patients with heart failure, there was no statistically significant correlation between pro- and antiangiogenic factors when comparing plasma and GCF. Other comorbidities, in most cases, do not show an impact on the change in general correlation for the entire group, as shown in Table 3. However, some of the groups were under-represented due to the percentages of comorbidities shown in Table 1.

The results obtained each time indicate a positive correlation with respect to a given parameter or do not show statistical significance. However, no negative correlations were observed, which would indicate a reversal of the trend of change in concentration of circulating pro- and antiangiogenic factors for the analyzed comorbidities. Patients with or without coronary artery disease have similar correlation trends for PDGF-BB concentration on days 1 and 30, with a *p*-value < 0.05.

## 4. Discussion

To our knowledge, there is not much research on the correlations of angiogenic factors between plasma and oral fluids. In our study, PDGF-BB was the biomarker with the most significant and positive correlation in plasma and GCF for the entire group (*p* < 0.017). Importantly, our results showed that the analyzed comorbidities did not influence the observed correlations. For PDGF-BB, positive correlations were observed for most studied conditions, which suggests its potential use also in heart failure, dyslipidemia, coronary arterial disease, and myocardial infarction. PDGF-BB is one of the isoforms of PDGF, representing a family of platelet-derived growth factors (PDGFs). PDGFs are expressed by various cell types, including vascular smooth muscle cells (VSMCs) and endothelium, stimulating mitosis in mesenchymal cells [16,17]. They have well-known angiogenic properties that depend on the type of endothelium [18,19]. However, in more recent studies, PDGF has gained recognition as one of the elements in the pathogenesis of cardiovascular disease. It plays an important role in the migration of VSMCs into the neointima after acute injury in atherosclerotic plaques [20]. Furthermore, the scientific literature is populated by reports describing PDGF inhibition’s effects on SMC proliferation and migration in atherosclerotic lesions. Therefore, substances with such properties are tested as anti-atherogenic therapies [21].

In the literature, PDGF-BB in GCF has been explored in terms of clinical parameters of periodontitis and in relation to nonsurgical periodontal treatment [22]. PDGF-BB was also compared after treatment of periodontal intrabony defects with guided tissue regeneration or access surgery [23]. However, a study that investigated PDGF levels in GCF in patients with coronary heart disease showed no difference compared to the control group. What is worth highlighting is that our conclusions contrast with those of this study, which showed no significant correlation between plasma and GCF for PDGF despite a similar number of analyzed subjects [24]. Therefore, more studies are required on a larger number of patients to investigate this significant finding due to the low impact of subgroups on quantitative measurements. Regarding PAD, several studies have been conducted on the determination of PDGF in various materials obtained from patients with chronic lower limb ischemia. A study carried out among subjects with PAD showed a reduction in circulating PDGF along with increased Rutherford stages [25]. A study by Ariyanti et al. conducted on individuals with DM2 and lower limb ischemia revealed that high blood sugar levels increase the production of prolyl hydroxylase domain 3 in skeletal muscle cells. This, in turn, leads to a reduction in the release of angiogenic factors, particularly VEGF-A and PDGF-BB [4]. These findings were supported by a rabbit hindlimb ischemia model, where the data presented revealed new evidence that shows the positive impact of DNA-based PDGF-BB on muscle recovery after ischemic injury [26].

The purpose of our publication was to assess the correlation between the individual diagnostic materials and the possibility of using an alternative material to plasma. However, based on the analysis, it was shown that concomitant diseases may affect the significance of changes in the concentration of circulating pro-angiogenic and anti-angiogenic factors over time, which may suggest the usefulness of assessing these parameters in individual disease entities. Due to the lack of publications in this area and the insufficient amount of clinical data, which does not allow for a thorough analysis in our study, it is advisable to conduct detailed studies to determine the usefulness of these determinations in particular diseases of the cardiovascular system.

Recent studies are increasingly emphasizing the use of GCF as a source of various predictive markers in diagnostics. The expression of miRNAs associated with cardiovascular disease (CVD) risk was investigated using GCF in patients with periodontal disease. It was found that periodontitis was a significant predictor of GCF miRNAs associated with CVD risk [27]. Furthermore, fluid from gingival pockets was examined as material for the evaluation of biomarkers in cardiovascular disorders such as atherosclerosis, arterial hypertension, coronary heart disease, or ischemic stroke [28]. In the pediatric population, researchers also utilized GCF as a means of identifying interleukin-1α (IL-1α) as a novel non-invasive marker for the detection of external inflammatory root resorption, which can occur following dental trauma [29]. In addition, apart from its traditional use in periodontal disease diagnosis, GCF gained attention as a potential source for assessing the oral cavity microbiome. The analysis of bacterial biofilms and the detection of viruses in the periodontal pocket using GCF samples is becoming one of the major interests of the scientific dental community [30,31]. One study indicated that GCF can even serve as a potential medium for evaluating the mycobiome, as it revealed the presence of *Aspergillus niger* in a patient with COVID-19 and could predict the occurrence of an invasive aspergillosis infection. These tests may be even faster and more sensitive than traditional serum examinations (including antigen tests) [32].

Regarding the evaluation of VEGF-A, our results suggested that a correlation between plasma and GCF concentrations occurred only among subjects with a smoking history and patients without heart failure. There were no significant correlations for the whole group. Despite scientific evidence of the influence of smoking on VEGF levels [33,34], it appears to have no impact on VEGF concentrations in GCF [35,36]. In a case-control study, the concentration of VEGF in blood samples was shown to be statistically higher in smokers than in non-smokers (*p* < 0.001) [37]. However, in systemically healthy individuals with optimum oral hygiene and a healthy periodontium, smoking alone was shown to have no significant impact on the evaluated growth factors in gingival pockets [35]. In our study, oral hygiene and dental status were not assessed, making it difficult to relate to the aforementioned findings in this context.

Our results revealed an inverse correlation between plasma HbA1c concentrations and VEGF-A levels in the GCF. This finding is in contrast with a previous study, where the concentrations of VEGF-A and its receptors in patients with well-controlled diabetes were comparable to those of healthy individuals. There were no correlations between VEGF-A and HbA1c in plasma [38]. In the study by Alimam et al., there was also a nonsignificant correlation between VEGF genotypes, HbA1c, and blood glucose levels ([39]).

Interestingly, in addition to the use of VEGF in the progression of periodontitis [40], it can also play an essential role as a biomarker in the progression of coronary heart disease. In the study by Haihua et al., patients were divided into three groups: periodontitis, coronary heart disease only, and periodontitis with coronary heart disease. The results showed that levels of VEGF in the GCF of all groups with the disease were significantly higher compared to the control group (with a significance level of (*p* ≤ 0.05), suggesting that VEGF can be considered a biomarker in the progression of coronary heart disease [24]. In general, its role in the pathogenesis of coronary heart disease is two-fold. VEGF-A promotes cardiomyocyte activation, leading to wound healing and morphogenesis. However, on the other hand, it may also be secreted by cardiac cells in response to inflammation, cytokine stimulation, or mechanical stress. What is important is that elevated levels of VEGF-A were detected in individuals suffering from various CVDs and are often linked to disease advancement and an unfavorable prognosis. Regarding the significance of VEGF among individuals with PAD, several studies showed different levels among patients with lower limb ischemia and healthy controls. It was shown that the VEGF-A/sVEGFR-1 (VEGF-A soluble receptor) ratio was reported to be higher in individuals with critical limb ischemia compared to subjects with intermittent claudication (IC) and a healthy group [41]. However, a systematic review evaluating current evidence of VEGF modulation in the context of PAD in animal models showed that VEGF-A positive modulation decreases lumen stenosis and neointimal hyperplasia [42]. In 2017, a systematic review on the therapeutic application of angiogenic factors in patients with PAD was published in the Cochrane Library. The results of this review showed that the utilization of growth factors such as VEGF may lead to improved blood flow measurements and a reduced risk of limb amputations, particularly in preventing minor amputations [3].

According to the circulating ANG-1 assessment, we found no positive correlations between its concentrations in plasma and oral fluids—even when the analysis was performed in subgroups. To date, ANG-1 in GCF has only been investigated as a biomarker for aggressive periodontitis and clinical outcomes after periodontal surgery [43]. However, several studies have shown that it may also play a role as a cardiovascular biomarker. Low circulating ANG-1 levels were shown to be positively associated with abnormal cardiac structure in patients with stages 3–5 of chronic kidney disease [44]. In addition, the concentration of ANG-1 in the coronary artery blood in patients with acute coronary syndrome was shown to be notably higher than that of the unstable angina pectoris group. There was also a positive correlation between ANG-1 concentration and the severity of coronary lesions. Moreover, patients who experienced recurrent major cardiovascular and cerebrovascular adverse events during follow-up were found to have higher levels of ANG-1 in the coronary sinus [45]. These findings are consistent with animal model studies, which demonstrated that ANG-1 may enhance the formation of atherosclerotic plaques and inhibit cholesterol efflux [46].

These findings are consistent with animal model studies that have shown that ANG-1 enhances the formation of atherosclerotic plaques and inhibits cholesterol efflux [47]. In current studies, potential salivary biomarkers have been discovered in such systemic diseases as autoimmune disorders, infections, malignancies, dental caries, periodontal diseases, and, increasingly, cardiovascular diseases [48]. Recent advances in highly sensitive technologies, including next-generation sequencing, highly sensitive ELISAs, mass spectrometry, and homogeneous immunoassays, have made it possible to accurately assay even small quantities of salivary biomarkers. This provides opportunities for the development of many future diagnostic applications. Oral fluids have been shown to produce more accurate, inexpensive, and convenient results compared to plasma. Saliva and GFC can be easily collected and readily used for tests. Furthermore, the collection of oral fluids poses a much lower risk of complications compared to blood sampling. This makes the method non-invasive and eliminates the need for specially trained personnel. It can also be applicable in pediatrics and among individuals who suffer from significant fear associated with blood collection. According to financial issues, one study showed that saliva collection was even 48% less costly than blood collection [49]. Our findings may become one of the reasons contributing to the increased use of newer, more accessible, and cheaper materials, such as those obtained from the oral cavity.

The main limitation of our incidence was the small number of patients tested, making it difficult to explore general directions and trends for research in a larger group. It also resulted in disproportions in demographic factors resulting from the demographics of patients reporting to the hospital with critical ischemia of the lower limbs, among whom men predominate. The data in the current literature are limited and do not allow for planning effective studies in larger groups to support our findings without prior testing in a small group due to the high cost of determining the angiogenic factors.

## 5. Conclusions

In summary, current data indicate that the appropriate use of biomarkers in patients with PAD and concomitant DM2 can contribute to an early diagnosis and even the subsequent improvement of current therapies. At the same time, research on the application of new biomarkers should be accompanied by studies on new diagnostic materials. Our results clearly showed that GCF could be a good source for the evaluation of PDGF and VEGF-A in some conditions that are not disturbed by the accompanying comorbidities. However, future studies with a larger number of subjects are warranted to confirm these findings and identify the most accurate angiogenic biomarkers in saliva or GCF that could be applied in clinical practice.

## Figures and Tables

**Figure 1 biomedicines-11-01596-f001:**
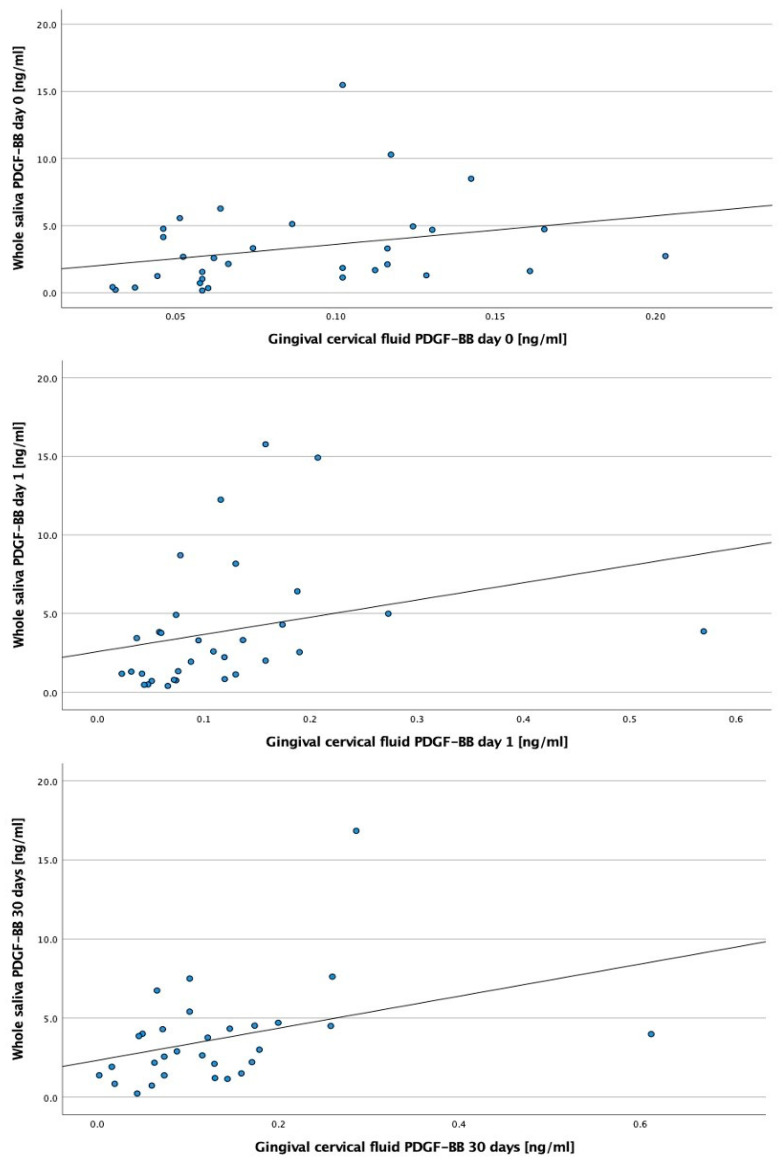
Scatter points of the PDGF-BB correlation between gingival crevicular fluid and whole saliva with a defined trend line.

**Table 1 biomedicines-11-01596-t001:** Characteristics of the study population.

Sex n (%) Males Females	27 (84.4%) 5 (15.6%)
Age median (IQR)	69.0 (62.5; 73.5)
Rutherford n (%) 4 5 6	2 (6.3%) 19 (59.4%) 11 (34.3%)
Hypertension n (%) No Yes	1 (3.1%) 31 (96.9%)
Myocardial infraction n (%) No Yes	22 (68.8%) 10 (31.3%)
Coronary artery disease n (%) No Yes	15 (46.9%) 17 (53.1%)
Smoking n (%) No Former Actual	7 (22.6%) 21 (67.7%) 3 (9.7%)
Heart failure n (%) No Yes	19 (59.4%) 13 (40.6%)
Dyslipidemia n (%) No Yes	7 (21.9%) 25 (78.1%)
Chronic kidney disease n (%) No Yes	25 (78.1%) 7 (21.9%)
Atrial fibrillation n (%) No Yes	25 (78.1%) 7 (21.9%)
ABI median (IQR)	0.5 (0.39; 0.72)
TBI median (IQR)	0.16 (0.07; 0.22)
Hgb median (IQR) (g/dL)	11.9 (11.05; 14.15)
WBC median (IQR) (10^3^/uL)	9.99 (8.43; 12.58)
CRP median (IQR) (mg/L)	6.98 (3.32; 59.25)
HbA1c median (IQR) (%)	7.45 (6.83; 8.13)

ABI—ankle-brachial index; CRP—c-reactive protein; HbA1c—glycated hemoglobin; Hgb—hemoglobin; TBI—toe-brachial index; WBC—white blood count.

**Table 2 biomedicines-11-01596-t002:** Median concentration of circulating proangiogenic and antiangiogenic factors at three time points across various study materials.

		Day 0	Day 1	Day 30
VEGF-A Median (IQR) (ng/mL)	plasma	0.054 [0.008; 0.129]	0.0613 [0.004; 0.099]	0.068 [0.004; 0.156]
	WS	1.804 [1.219; 2.069]	1.501 [1.219; 1.748]	2.010 [1.679; 2.199]
	GCF	0.102 [0.038; 0.171]	0.072 [0.029; 0.148]	0.110 [0.038; 0.190]
PDGF-BB Median (IQR) (ng/mL)	plasma	0.237 [0.119; 0.475]	0.256 [0.116; 0.460]	0.295 [0.151; 0.451]
	WS	0.005 [0.004; 0.007]	0.005 [0.003; 0.007]	0.004 [0.003; 0.007]
	GCF	0.007 [0.006; 0.012]	0.009 [0.006; 0.015]	0.011 [0.006; 0.017]
ANG-1 Median [IQR] (ng/mL)	plasma	1.383 [0.378; 2.588]	1.215 [0.448; 2.126]	1.443 [0.966; 1.997]
	WS	2.879 [1.836; 3.294]	2.694 [1.891; 3.203]	3.106 [2.291; 3.897]
	GCF	0.020 [0.012; 0.034]	0.021 [0.008; 0.050]	0.024 [0.015; 0.045]

**Table 3 biomedicines-11-01596-t003:** Correlation analysis for serum versus GCF in three-point time analysis, including the presence or lack of comorbidity.

Analyzed Comorbidity	Correlated Factors	R	*p*-Value
No Heart Failure	VEGF1	0.12	0.63
	VEGF2	0.35	0.15
	**VEGF3**	**0.59**	**0.013**
	**PDGF1**	**0.46**	**0.047**
	**PDGF2**	**0.58**	**0.009**
	PDGF3	0.45	0.068
Former smoking	VEGF1	0.39	0.08
	**VEGF2**	**0.47**	**0.031**
	**VEGF3**	**0.50**	**0.029**
	PDGF1	0.32	0.15
	**PDGF2**	**0.44**	**0.049**
	**PDGF3**	**0.54**	**0.017**
Dyslipidemia	**PDGF1**	**0.421**	**0.036**
	**PDGF2**	**0.62**	**0.001**
	PDGF3	0.37	0.085
No coronary artery disease	PDGF1	0.50	0.056
	**PDGF2**	**0.62**	**0.013**
	**PDGF3**	**0.57**	**0.035**
Coronary artery disease	PDGF1	0.31	0.224
	**PDGF2**	**0.49**	**0.047**
	**PDGF3**	**0.56**	**0.025**
No myocardial infraction	**PDGF1**	**0.58**	**0.005**
	**PDGF2**	**0.49**	**0.021**
	**PDGF3**	**0.55**	**0.012**
Myocardial infarction	PDGF1	0.12	0.73
	**PDGF2**	**0.65**	**0.042**
	PDGF3	0.53	0.12

Bolded significant results.

## Data Availability

The datasets generated and analyzed during the current study are available from the corresponding author on reasonable request.

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
