# Peer review of "The Concentration of Pro- and Antiangiogenic Factors in Saliva and Gingival Crevicular Fluid Compared to Plasma in Patients with Peripheral Artery Disease and Type 2 Diabetes"

_biomedicines, 2023, doi:10.3390/biomedicines11061596_

Round 1

Reviewer 1 Report

Research article by Iwona Gregorczyk-Maga et al, entitled ‘The concentration of pro- and antiangiogenic factors in saliva and gingival crevicular fluid compared to plasma in patients with peripheral artery disease’ deals with one of the important diseases of CV. This study is more focused on diagnostic purposes rather than the biomarkers for PAD. The author’s attempt was good to establish an easy way to diagnose PAD. I have some questions related to this manuscript before my final decision.

Authors should simplify the following sentence for the general audience ….’Saliva is a plasma ultrafiltrate comprising proteins synthesized in situ from blood or in the salivary glands (11)’. Similar sentences should be corrected for structure and meaning.

Please justify time points selection 0, 1, and 30 days. While all the patients have PAD. What is the conclusion of this time point's results?

The sample size is too small as per the diagnostic study. On top of this authors split data on the basis of subsets, further decreasing the ‘n’.

Are all patients assumed to have type 2 diabetes.???? Please provide data in the manuscript. Glucose or HBA1c.

Check for typo errors like Miocardial etc…….

The authors should mention whether any of the participants had oral disease or issues at the time of the study or before. (Or overall, any infectious disease). Oral diseases could affect these markers.

Why authors did not study the VEGF isoforms since VEGF isoforms have different roles in angiogenesis?

Is this data adjusted to any parameter?

Please mention the ELISA cutoff value for these markers in the method section. Also, the way ELISA is performed like dilution of the samples, etc. Authors should discuss whether these diagnostic tests are noninvasive etc. Also, the benefit of this method over serum/plasma samples estimation.

Authors should cite data from the study that also included control subjects.

Authors should add more scientific literature data related to PAD and VEGF, PDGF levels from different body sources, or even from animal models to strengthen the biomarker part of the study.

The discussion section is too exhausting. Discussion should be aligned with the study findings.

Data analysis and legends could be better.

Also, I feel this manuscript should be written in a compact way to understand it well.

Overall moderate editing is required.

Author Response

We are grateful for any suggestions to improve our manuscript. We have made every effort to meet all comments. I invite you to read the attached file containing the answers according to the order of comments.

Reviewer 2 Report

The manuscript studied the levels of angiogenic factors in saliva and gingival crevicular fluid (GCF) compared to blood in diabetic patients with saliva and gingival crevicular fluid (GCF). The authors demonstrated that that GCF could be a good source for the evaluation of PDGF and VEGF-A in some conditions

-The patients included in the study are all diabetic, this should be indicated in title, conclusion and discussion.

-The study has many limitations. The sample size and the greater number of males compared to female

-Presentation of results is confusing. It is suggested to present results as ng/ml instead of pg/ml. Also, the use of decimal point and/or comma for numbers should be revised

The English language needs moderate revision

Author Response

(The authors gave the same response as above.)

Round 2

Reviewer 1 Report

Authors reply to all the comments.

The manuscript still faces formatting and sentence construction problems in some places. I found it in the abstract initially.

Overall I accept this manuscript. The rest authors need to improve formatting and sentence construction. I hope the editorial will help regarding this issue.

Authors could use Grammarly software. It's just a recommendation because many people including Prof. use it nowadays.

Formatting and sentence construction 

Author Response

We sincerely appreciate your comments to improve our manuscript. According to the editor's proposal, the manuscript is submitted for editorial editing and minor language correction by the editor.

Reviewer 2 Report

The authors addressed my comments

English language is ok

Author Response

We sincerely appreciate once again for your comments to improve our manuscript.